# *Arsenophonus*: A Double-Edged Sword of Aphid Defense against Parasitoids

**DOI:** 10.3390/insects14090763

**Published:** 2023-09-13

**Authors:** Minoo Heidari Latibari, Gholamhossein Moravvej, Ehsan Rakhshani, Javad Karimi, Diana Carolina Arias-Penna, Buntika A. Butcher

**Affiliations:** 1Department of Plant Protection, Faculty of Agriculture, Ferdowsi University of Mashhad, Mashhad P.O. Box 91779-48974, Iran; minoo.heidarilatibari@mail.um.ac.ir (M.H.L.); jkb@um.ac.ir (J.K.); 2Department of Plant Protection, Faculty of Agriculture, University of Zabol, Zabol P.O. Box 538-98615, Iran; rakhshani@uoz.ac.ir; 3Unaffiliated Entomologist, Bogotá D. C. 111221, Cundinamarca, Colombia; cdianaa@gmail.com; 4Integrative Insect Ecology Research Unit, Department of Biology, Faculty of Science, Chulalongkorn University, Bangkok 10330, Thailand

**Keywords:** biological control, black cowpea aphid, parasitoid, secondary symbiont

## Abstract

**Simple Summary:**

Endosymbiont interactions with hosts have significant effects on pests and beneficial species. A form of endosymbiosis is known as a mutually beneficial association. In this context, specific facultative endosymbiotic bacteria, such as *Arsenophonus*, play a defensive role by safeguarding aphids against parasitoids. This study on black cowpea aphids (BCAs) revealed that *Arsenophonus* does not prevent parasitism by two species of Aphidiinae wasps, *Binodoxys angelicae* and *Lysiphlebus fabarum*. However, the maturation and emergence of adult *B. angelicae* wasps were delayed when BCAs were parasitized. This delay limits the effectiveness of *B. angelicae*, reducing the number of emerged adult wasps and allowing other members of the BCA colony to survive. The role of *Arsenophonus* in the interaction between *A. craccivora* and its parasitoids is multifaceted and acts as a double-edged sword.

**Abstract:**

It is widely accepted that endosymbiont interactions with their hosts have significant effects on the fitness of both pests and beneficial species. A particular type of endosymbiosis is that of beneficial associations. Facultative endosymbiotic bacteria are associated with elements that provide aphids with protection from parasitoids. *Arsenophonus* (Enterobacterales: Morganellaceae) is one such endosymbiont bacterium, with infections being most commonly found among the Hemiptera species. Here, black cowpea aphids (BCAs), *Aphis craccivora* Koch (Hemiptera: Aphididae), naturally infected with *Arsenophonus,* were evaluated to determine the defensive role of this bacterium in BCAs against two parasitoid wasp species, *Binodoxys angelicae* and *Lysiphlebus fabarum* (both in Braconidae: Aphidiinae). Individuals of the black cowpea aphids infected with *Arsenophonus* were treated with a blend of ampicillin, cefotaxime, and gentamicin (*Arsenophonus*-reduced infection, AR) and subsequently subjected to parasitism assays. The results showed that the presence of *Arsenophonus* does not prevent BCAs from being parasitized by either *B. angelicae* or *L. fabarum*. Nonetheless, in BCA colonies parasitized by *B. angelicae*, the endosymbiont delayed both the larval maturation period and the emergence of the adult parasitoid wasps. In brief, *Arsenophonus* indirectly limits the effectiveness of *B*. *angelicae* parasitism by decreasing the number of emerged adult wasps. Therefore, other members of the BCA colony can survive. *Arsenophonus* acts as a double-edged sword, capturing the complex dynamic between *A. craccivora* and its parasitoids.

## 1. Introduction

The majority of insects and most other eukaryotic organisms carry bacteria that have symbiotic relationships with them [1]. An example of this interaction involves beneficial (mutualistic) associations where bacteria reside within the host’s body (e.g., within tissues or cells). These endosymbiont bacteria can provide essential nutrients, vitamins, and other compounds that their hosts are incapable of producing themselves or even help them improve their food intake [2]. These are the so-called obligate symbionts because of their vital role in the host’s survival. These bacteria are transferred from the mother to the offspring during reproduction, ensuring their passage from one generation to the next [3]. In contrast, facultative symbionts are not required to be present in every individual of a host population for the host’s survival. Thus, facultative symbionts may affect the host´s fitness in three different ways: beneficially (positively), detrimentally (negatively), or neutrally (no effect) [1]. In addition, they can spread across the host populations through the process of distorting reproduction (sex-ratio distorters are heritable elements that modify the sex ratio of their host to promote their transmission), making it more likely for the symbiont to be passed on maternally via eggs [4]. These bacteria secrete chemical compounds to their host that offer benefits such as increasing the host’s resistance to natural enemies or improving their tolerance to environmental stressors [5,6,7]. Moreover, the bacteria are capable of supplying the host with nutrients, which facilitate rapid growth and development [8]. These facultative symbionts are of particular interest due to their capability for horizontal inter- and intra-specific transmission, which leads to the instantaneous acquisition of beneficial traits [9].

Since aphids have a wide range of symbiotic relationships with a variety of bacteria [10], they are reliable models to investigate nutritional and protective symbioses. As these relationships are strictly regulated, they offer insight into the mechanisms of symbiosis [11]. Almost all aphid species harbor obligate symbionts that provide essential amino acids and nutrients that the aphid cannot synthesize on its own. Therefore, the presence of the symbiont is crucial for the host’s survival [12]. Additionally, the majority of aphids are infected with at least one facultative maternally transmitted symbiont [13].

*Arsenophonus* (Enterobacterales: Morganellaceae) is a Gram-negative bacterium that forms facultative symbiotic relationships with approximately 5% of arthropods host [14]. The bacterium specializes in suppressing parasitoids and pathogens, allowing the hosts to better survive in their environment [15]. However, the functioning of *Arsenophonus* in these hosts remains unclear. It has been suggested that *Arsenophonus* could operate as a defense mechanism against parasites or other invading organisms. This defensive capability could be leveraged to help manage and reduce pest populations, potentially resulting in more efficient pest control solutions [16].

*Arsenophonus* is found in multiple crop-damaging aphids, including the black cowpea aphid (BCA) and *Aphis craccivora* Koch (Hemiptera: Aphididae) [17]. This aphid species is a major pest in alfalfa fields. It causes direct damage by sucking the sap from the plants [18] or indirect damage by transmitting viruses that can reduce yields and stunt the growth of plants (e.g., Alfalfa enation virus (AEV) and Alfalfa mosaic virus (AMV)) [19,20]. Black cowpea aphids possess multiple heritable symbiotic bacteria, making them some of nature´s more notorious pests [21].

Aphid populations can be biologically controlled through parasitoids, small insects that live at the host’s expense and eventually cause their death [22,23,24]. Aphidiinae, a subfamily of Braconidae parasitoid wasps, specialize in using aphids as hosts, therefore reducing the aphid population [20,25].

*Binodoxys angelicae* (Haliday) is a common Aphidiinae parasitoid species in the Middle East used as a biological control agent against aphids in agriculture and horticulture [26]. It can target more than 30 aphid species as hosts, including the green peach aphid (*Myzus persicae* Sulzer), soybean aphid (*Aphis glycines* Matsumura), and black cowpea aphid, which are major pests on crops like cereals, fruits, vegetables, and ornamental plants [27]. Another Aphidiinae parasitoid wasp is *Lysiphlebus fabarum* (Marshall). This species, found in countries such as Iran, is effective in controlling aphid populations [28] and has a broad host range, with over 40 aphid species reported as hosts. Some of the aphid species targeted by *L. fabarum* are aphids, which are the main pests on fruits, vegetables, and ornamental plants [27].

The outcome of insect–symbiont interactions may influence the effectiveness of biological control programs and the performance of insect hosts [29]. Recent parasitism surveys reported that the BCA can be parasitized by several species of Aphidiinae wasps [30,31] and infected by different facultative symbionts, such as *Arsenophonus* [32]. This infection is beneficial to the aphid as it provides protection against its natural enemies [33]. Nevertheless, there is evidence indicating the opposite: *Arsenophonus* infections may not necessarily be beneficial for their hosts [16].

So far, only *Arsenophonus,* a facultative endosymbiotic bacterium, has been detected in the populations of *A. craccivora* across alfalfa sampling stations in the northeast and northwest regions of Iran, based on the 16S ribosomal RNA (rRNA) metabarcoding analysis. Here, the *Arsenophonus* infection status of *A. craccivora* feeding on alfalfa fields was evaluated. In addition, it was investigated whether the presence of *Arsenophonus* in BCAs protects them against two parasitoid wasps, *L. fabarum* and *B. angelicae*.

## 2. Materials and Methods

### 2.1. Sampling Sites

Sampling was conducted during the typical alfalfa growth period in the northeastern region of Iran, which extended from mid-March to mid-October 2019. Black cowpea aphids and their parasitoids were collected from three alfalfa farms near the Mashhad metropolis in Razavi Khorasan Province. These sites were: (a) Toroq, 36°12′39″ N, 59°39′01″ E, 1007 m; (b) Hesar-e Sorkh, 36°24′52″ N, 59°20′38″ E, 1250 m; and (c) Kahu, 36°27′08″ N, 59°13′30″ E, 1370 m (Figure 1).

### 2.2. Rearing Specimens

#### 2.2.1. Aphids

A total of eight black cowpea aphid (BCA) colonies (I–VIII) were established from individuals collected from randomly selected alfalfa leaves (*Medicago sativa* L., Fabales: Fabaceae). Thus, colonies I–III were derived from site a, colonies IV–V from site b, and colonies VI and VIII from site c. The presence of *Arsenophonus* in all eight colonies was evaluated by microbiome observations (see details below). For this experiment, colony I, which reported the lowest level of *Arsenophonus* infection, was excluded.

#### 2.2.2. Rearing of BCAs from the Colonies II–VIII

To initiate female lines, young female foundresses were released individually onto alfalfa pots in a growth chamber and reared at a temperature of 21 ± 1 °C, with a relative humidity (RH) of 60% ± 5%, and a photoperiod of 16L:8D. To prevent overcrowding and alate production, asexual wingless females were transferred to new alfalfa plants as needed (approximately monthly) [16]. All BCAs were maintained under the same conditions from the established colonies until the experiments. The experimental plants were enclosed in mesh-covered cylindrical cages (35 × 20 cm; height × diameter) [20].

#### 2.2.3. Parasitoids

Adults of *Lysiphlebus fabarum* and *Binodoxys angelicae* were the most common parasitoid wasp species. These species were obtained from locally collected BCAs that had been parasitized in the alfalfa fields. Parasitized aphids (detected by the formation of mummified aphids) were brought to the laboratory and reared in a controlled environment room at 21 ± 1 °C, 60% ± 5% RH, and 16L:8D. Three wild-caught parasitized BCAs were placed individually on an alfalfa leaf in a glass dish over a layer of moist filter paper. Female adult wasps were then fed honey and water after emergence. Newly emerged BCAs with *Arsenophonus*-reduced infection (AR) and growing on *M. sativa* were used for establishing isofemale lines of parasitoids (see below). Identifications of the emerged wasps were confirmed using taxonomic resources [34]. At least three generations of parasitoids were reared prior to starting the assays. Both parasitoid species and BCAs were reared at the laboratory and experimental greenhouse of the Department of Plant Protection, Ferdowsi University of Mashhad, Iran.

### 2.3. Species Identification

The alfalfa plant and the BCA species were determined following the taxonomic keys by Lesins and Lesins (1979) and Blackman and Eastop (2017) [35,36], respectively. Identifications of *B. angelicae* and *L. fabarum* adults were generated by integrating two datasets: relevant external morphological traits [34] and mitochondrial cytochrome oxidase subunit I (COI) gene sequence data [37,38]. COI sequences are deposited in GenBank (http://www.ncbi.nlm.nih.gov/genbank/, accessed on 5 September 2023).

### 2.4. Images

External morphology traits for parasitoid wasps were examined with an Olympus^®^ SZX9 (Olympus Corporation, Tokyo, Japan). The larva of *B. angelicae* was digitally photographed using a Dino-lite digital eyepiece camera and a conventional light diffuser, and the resulting images were further processed with Adobe Photoshop^®^ CS6.

### 2.5. Arsenophonus-Reduced Infection (AR)

The second-instar nymphs of the BCAs clone were obtained from colonies II–VIII. Nymphs were reared on their natural hosts and fed artificial diets with antibiotics (McLean, pers. comm.). A blend of three commonly used antibiotics, ampicillin, cefotaxime, and gentamicin, was utilized to target Gram-negative bacteria [39]. *Medicago sativa* leaves were placed in Eppendorf tubes. These tubes contained a solution of ampicillin (100 mg·mL^−1^), cefotaxime (50 mg·mL^−1^), and gentamicin (50 mg·mL^−1^) (Tikochem Company, Tehran, Iran). Then, the stalk of each cut leaf was submerged in the solution, and the gap between the stalk and the edge of the tube was sealed with parafilm. Second-instar nymphs of aphids were then allowed to feed on the treated leaves for a period of 3 to 4 days in a controlled environment room at 14 °C. 

Some bacterial strains were more difficult to eliminate than others, and a single round of antibiotics was insufficient to eliminate the bacteria. Thus, after two days of rest to recover from the initial infection, the aphids were re-treated with antibiotics [40]. Then, the BCAs were placed on individual alfalfa leaves and monitored for survival. After six generations, a subset of offspring was tested for the presence or absence of *Arsenophonus* via polymerase chain reaction (PCR) experiments before and after the use of antibiotics. These diagnostic PCR tests (standard PCR) were conducted to examine any changes or effects caused by the antibiotics on the status of natural *Arsenophonus* infection in aphids.

### 2.6. Molecular Methods

The DNA of *L. fabarum* and *B. angelicae* was extracted using entire female wasps. BCAs were carried out using a combination of the whole body of the aphid nymphs at different stages (three in the second instar plus two in the third instar). DNA was extracted using the RNA/DNA Geneaid™ Blood and Tissue Kit (Geneaid™ Biotech Ltd., New Taipei City, Taiwan) according to the manufacturer’s instructions. All DNA samples were electrophoresed in a 0.10% agarose gel and visualized under a UV transilluminator to observe DNA that has been separated into bands by electrophoresis through an agarose gel. DNA concentrations were standardized to 50 ng/mL and stored at −20 °C until PCR analysis. For the parasitoid specimens, the COI fragment (603 bp) was amplified using universal primer pairs, LCO1490 (GGTCAACAAATCATAAAGATATTGG) and HCO2198 (TAAACTTCAGGGTGACCAAAAAATCA) [37]. The PCR tests used to identify parasitoids were carried out twice. The presence/absence of *Arsenophonus* before and after antibiotic use was tested based on the ftsK (filament temperature-sensitive mutant K) and fbaA (fructose-bisphosphate aldolase class II) genes in six independent diagnostic PCR experiments using positive control, negative control, and PCR products, all with *Arsenophonus*-specific primers. The presence of *Arsenophonus* was detected using two primer sets: ftsK [ftsKf (GTTGTYATGGTYGATGAATTTGC) and ftsKr (GCTCTTCATCACYATCAWAACC)] and fbaA [fbaAf (GCYGCYAAAGTTCRTTCTCC), fbaAr2 (GGCAAATTAAATTTCTGCGCAAC G)] [41].

All PCR reactions were performed in total volumes of 14 μL by using 1 μL of DNA template, 5× buffer, 10 mM of each dNTP, 10 mM of each primer, and 1.25 u/mL DNA polymerase. For parasitoids, PCR thermal cycling conditions consisted of an initial denaturation at 94 °C for 5 min, followed by 30 cycles of 95 °C for 40 s, 50 °C for 30 s, and 72 °C for 40 s, followed by an extension at 72 °C for 10 min. The reactions were held at 16 °C. For *Arsenophonus*, PCR amplifications were performed under the following conditions: initial denaturation at 93 °C for 3 min, 30 cycles of denaturation (93 °C, 30 s), annealing (50–52 °C, depending on primers, 30 s), extension (72 °C, 1 min), and a final extension at 72 °C for 5 min [41]. PCR products from parasitoids and BCAs were electrophoresed in 1% agarose gel, and the manufacturer’s protocol (Geneaid™ kit) was used to clean PCR products. Parasitoid amplicons were sent to Macrogen (Seoul, Republic of Korea) for sequencing.

### 2.7. Parasitism Assay

The parasitism rate of *B. angelicae* and *L. fabarum* was assessed on *A. craccivora* under two conditions: (a) when infected with *Arsenophonus* and (b) when treated with a blend of three antibiotics (*Arsenophonus*-reduced infection (AR)). Thus, the control group (CG) consisted of *B*. *angelicae* parasitizing *A. craccivora* obtained from colonies II–VIII with the highest level of *Arsenophonus* infection (40 replicates). The experimental group comprised *B. angelicae* parasitizing *A. craccivora* with *Arsenophonus*-reduced infection (AR) (40 replicates). In each group, alfalfa stems (10 cm long) with leaves highly infested with nymphs (100 specimens in the second and third instars) were transferred to plastic containers (20 cm × 15 cm × 5 cm). The container (10 × 5 cm) was covered with mesh. The introduction of one set comprising a newly emerged female and male of *B. angelicae* to each container was followed by removing the parasitoids after 24 h. The BCA individuals exposed to parasitoids were reared under experimental conditions using alfalfa leaf disks until mummies appeared. It was verified that no other parasitoid or hyperparasitoid species were present. Parasitized aphids (mummies) were counted, and the parasitism rate was calculated (the number of mummies observed divided by the initial aphid number). Additionally, for each group (control + experimental), we calculated the following parameters: the number of days it took for the parasitoids to reach the pupal stage (the aphid body turns into a mummy), the number of days it took for the parasitoids to emerge as adults, the number of emerged adult wasps, and the rate of emerged adults (the number of emerged adults divided by the initial number of aphids). All the steps of this parasitism assay were performed exactly for *L. fabarum*. In the end, there were a total of 160 experiments: the control group with 80 replicates and the experimental group with 80 replicates.

### 2.8. Analysis

The Basic Local Alignment Search Tool (BLAST, https://blast.ncbi.nlm.nih.gov/Blast.cgi, accessed on 5 September 2023) was used to find regions of local similarity between the nucleotide sequences (COI) obtained from the two parasitoid species. Assembly of the forward and reverse sequences was performed prior to sequence registration. The nucleotide sequences are deposited in the GenBank database (http://www.ncbi.nlm.nih.gov/genbank/, accessed on 5 September 2023).

To assess whether there were significant differences in parasitism rates between *B. angelicae* and *L. fabarum* on *A. craccivora* infected with *Arsenophonus* and *A. craccivora* with *Arsenophonus*-reduced infection (AR), a parametric *t*-test (*p* < 0.05) was conducted, assuming independence, normal distribution, and homogeneity of variance. The data were analyzed using the R Core Team [42]. The *t*-tests were performed with the “nlme” package, while independent *t*-tests were conducted with the “tidyverse” package. Additionally, visual diagrams were generated using the “ggplot2” package to illustrate the analysis results.

## 3. Results

The nucleotide sequence in BLAST analyses confirmed the taxonomic identification for both parasitoid species, *B. angelicae* (accession number: OR262501) and *L. fabarum* (accession number: OR484804).

### 3.1. BCAs Treated with an Antibiotic Mixture (Cured Aphids)

In the initial round of using antibiotics aiming to reduce *Arsenophonus* in aphids, the brightness of the *Arsenophonus* band observed in the cured aphids (*Arsenophonus*-reduced infection—AR) was significantly diminished when compared to the control group (size of marker’s bands: ftsK: 400 bp, fbaA: 573 bp), although not entirely eliminated. However, in the second round of antibiotics, the cured aphids tested negative for *Arsenophonus*, indicating their complete absence. The accomplishment of eliminating *Arsenophonus* from the aphids was indicated by the absence of any bands for this bacterium when using the specific primers (Figure 2).

### 3.2. Parasitism Assays

#### 3.2.1. Number of Mummified BCAs (Parasitism Rate) (Figure 3A and Figure 4A)

The number of mummified BCAs infected with *Arsenophonus* did not show a significant difference when compared to BCAs with *Arsenophonus*-reduced infection in terms of parasitism by *B. angelicae* (*t* = −0.94, df = 78, *p* = 0.34) and *L. fabarum* (*t* = −0.50, df = 78, *p* = 0.61).

**Figure 3 insects-14-00763-f003:**
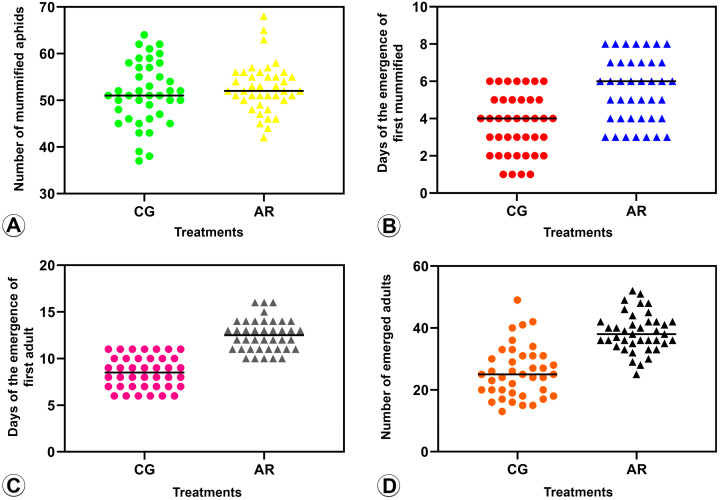
Parasitism assay of *Binodoxys angelicae* (Braconidae: Aphidiinae) on *Aphis craccivora* (Hemiptera) under two conditions: (a) infected with *Arsenophonus* (CG) (dots) and treated with a blend of three antibiotics (AR) (triangles). (**A**) Number of mature wasp larvae, as evidenced by mummified aphids, (AR) ^ns^. (**B**) Time (in days) it took for the wasp to become a larva, as evidenced by mummified aphids (AR) **. (**C**) Time (in days) it took for the wasp to emerge as an adult (AR) **. (**D**) The number of adult wasps that emerged (AR) **. ^ns^ = not significant; ** = significant difference between treatments.

**Figure 4 insects-14-00763-f004:**
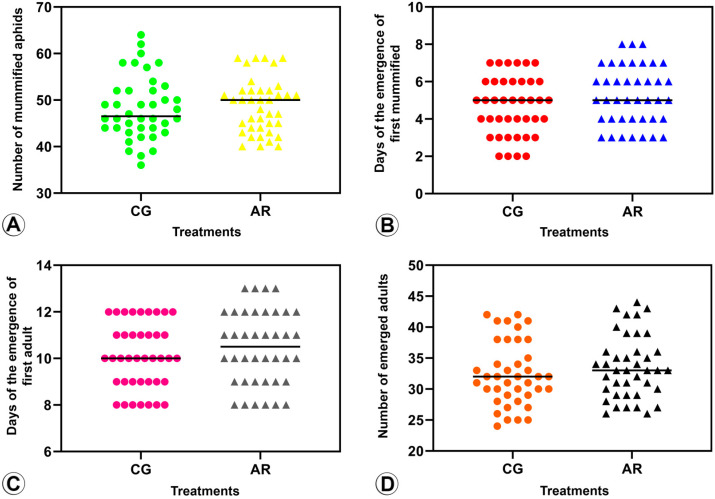
Parasitism assay of *Lysiphlebus fabarum* (Braconidae: Aphidiinae) on *Aphis craccivora* (Hemiptera) under two conditions: (a) infected with *Arsenophonus* (CG) (dots) and treated with a blend of three antibiotics (AR) (triangles). (**A**) Number of mature wasp larvae, as evidenced by mummified aphids, (AR) ^ns^. (**B**) Time (in days) it took for the wasp to become a larva, as evidenced by mummified aphids (AR) ^ns^. (**C**) Time (in days) it took for the wasp to emerge as an adult, (AR) ^ns^. (**D**) Number of adult wasps that emerged (AR) ^ns^. ^ns^ = not significant.

#### 3.2.2. Pre-Imaginal Period (Figure 3B and Figure 4B) 

The days to observe the mummified BCAs were statistically significant for *B*. *angelicae.* This indicates that it took longer for *B. angelicae* to reach the stage of a mature larva (evidenced by the presence of mummified aphids) in BCAs with *Arsenophonus*-reduced infection (AR) compared to BCAs infected with *Arsenophonus* (*t* = −4.94, df = 78, *p* = 0.001). However, there was no significant difference in the number of days it took to observe the mummified BCAs exposed to *L. fabarum* between BCAs infected with *Arsenophonus* and BCAs with *Arsenophonus*-reduced infection (AR) (*t* = −1.64, df = 78, *p* = 0.10).

#### 3.2.3. Period of Mummification to Adult Emergence (Figure 3C and Figure 4C)

The emergence time for *B. angelicae* as an adult was statistically significant, meaning that it took more days for this species to emerge as an adult on BCAs with *Arsenophonus*-reduced infection (AR) compared to BCAs infected with *Arsenophonus* (*t* = −10.44, df = 78, *p* = 0.001). However, there was no significant difference in the number of days it took for *L. fabarum* to emerge as adults between BCAs infected with *Arsenophonus* and BCAs with *Arsenophonus*-reduced infection (AR) (*t* = −1.04, df = 78, *p* = 0.30).

#### 3.2.4. Emergence Rate (Figure 3D and Figure 4D)

For *B. angelicae*, the number of emerged adults was significantly higher in BCAs with *Arsenophonus*-reduced infection (AR) than in BCAs infected with *Arsenophonus* (*t* = −7.80, df = 78, *p* = 0.001). In contrast, for *L. fabarum*, there was no significant difference in the number of emerged adults between BCAs infected with *Arsenophonus* and BCAs with *Arsenophonus*-reduced infection (*t* = −1.003, df = 78, *p* = 0.319).

It is noteworthy that for *B. angelicae*, the formation and number of mummies and adult parasitoids were lower in BCAs infected with *Arsenophonus* than in BCAs without Arsenophonus (AR). Because of that, supplementary observations were conducted. It was observed that on BCAs infected with *Arsenophonus*, several mummified aphids did not emerge as adults, resulting in a reduction in the total number of adults that did emerge. It was observed that the majority of the eggs of the parasitoid advanced to the later larval stage, but they were unable to develop into pupae or adults. As a result, the facultative endosymbiont *Arsenophonus* displayed a delayed defensive response when protecting BCAs against *B. angelicae* (Figure 5).

## 4. Discussion

Symbionts have numerous effects on their host aphids, including resistance to heat shock, parasitoids, and fungi [43]. The endosymbiotic relationship between *Arsenophonus* and aphids serves as an example of the intricate web of interactions that exist within the natural world. Through their coexistence, *Arsenophonus* and aphids have forged a remarkable partnership. Recent studies suggest that *Arsenophonus* can play a defensive role in protecting its aphid hosts against natural enemies such as parasitoid wasps and fungal pathogens [12,44]. Two defense mechanisms have been proposed. The first one is that *Arsenophonus* induces changes in the nutritional content of the aphid’s hemolymph (e.g., increasing the levels of certain amino acids [13]), making it less suitable for the parasitoid wasp larvae’s development, leading to their slower growth and development [45]. These findings lend support to the notion that the bacterium acts as a protective mechanism against wasps, as the compromised quality of the hemolymph poses challenges to the survival and reproductive success of wasp larvae. By decreasing the hemolymph quality, the presence of the bacterium creates a hostile environment that hinders the development and proliferation of wasps, further strengthening the defense hypothesis. The parasitoid wasp consumes the internal tissues and fluids of its host during the larval stage [46], thus having the greatest impact on the survival and reproduction of the aphid host. This intricate relationship between the bacterium, hemolymph quality, and wasp larvae underscores the complex dynamics at play in the ongoing battle between aphids and their natural parasitoids. Studies have suggested that aphids infected with *Arsenophonus* can increase their resistance to parasitoid wasps during the larval stage [47] rather than during the initial stage of the development of the parasitoid, the egg stage [48]. The second mechanism is that *Arsenophonus* alters the behavior of the parasitoid wasp, resulting in a decreased probability of parasitizing the aphid host [49]. It has been reported that *Arsenophonus* induces chemical changes in the aphid host cuticle, which can repel the parasitoid wasp [50]. This suggests that *Arsenophonus* is able to manipulate the behavior of the aphid by altering its cuticle composition, which can interfere with the parasitic performance of the wasp. Additionally, the protection provided by *Arsenophonus* varies depending on the type of insect host and the natural enemy involved [51].

Reduction or removal of facultative symbionts from aphid hosts can delay both the development of the mature larvae (evidenced by mummified aphids) and the emergence of adult parasitoid wasps [12,52,53,54]. This delay would be associated with the reduced nutritional quality of the aphid host’s hemolymph, which, in turn, hinders the development of the parasitoid wasp [45,55]. Here, the facultative symbiont *Arsenophonus* did not have a significant effect on the parasitism rate or the emerged adults of *L. fabarum* parasitizing BCAs. A possible explanation is that *Arsenophonus* is not effective in the development and survival of certain species of *Lysiphlebus* larvae [56,57].

On the other hand, our findings also show that in the case of BCAs parasitized by *B. angelicae*, the presence of *Arsenophonus* has significant effects on the development and emergence of both the parasitized aphid larvae and the adult parasitoid wasps. When aphids are parasitized by *B. angelicae*, the normal progression of larval maturation is disrupted. Additionally, the presence of *Arsenophonus* extends the duration of the maturation period for the parasitized aphid larvae. This delay in maturation can be attributed to the intricate interactions among the aphid host, specific parasitoid wasp species, and endosymbiont strain [16]. Furthermore, the presence of *Arsenophonus* causes a significant delay in the emergence of adult parasitoid wasps.

*Arsenophonus* seems to influence the development and behavior of the parasitoid wasps, causing significant delays in their emergence as adults. These delays indirectly impede the overall performance and effectiveness of *B. angelicae* in reducing aphid infestations, an issue not previously investigated [16]. These results highlight the intricate interplay among the host, the parasitoid, and the endosymbiont in this complex ecological relationship [58].

Overall, our results suggest that *Arsenophonus* is unlikely to serve as a defensive symbiont in alfalfa aphid. However, it is important to note that other genotypes of alfalfa aphid hosts may receive protection from different strains of *Arsenophonus*. This is demonstrated by the fact that various strains of *Hamiltonella defensa*, another bacterial endosymbiont, provide differing levels of protection against parasitism in pea aphids based on the presence, absence, and type of APSE (*Acyrthosiphon pisum* Secondary Endosymbiont) phage [50]. Furthermore, it has been revealed that a strain of *Regiella insecticola* possesses the ability to defend its aphid host against parasitism, a trait previously not attributed to this particular bacterial endosymbiont [59]. These findings highlight the fact that bacterial strains can exhibit unique defensive properties and emphasize the potential variability in the defensive capabilities of different strains.

## 5. Conclusions

There is a complex and multifaceted relationship between aphids and their facultative symbiont, *Arsenophonus*. In some cases, *Arsenophonus* enhances aphids’ defense response against parasitoids. However, the other way around has also been reported: *Arsenophonus* can make aphids more vulnerable to parasitoid attacks by changing their behavior or interfering with their immune system. In essence, *Arsenophonus* is considered a double-edged sword, as its presence may or may not be effective on its host’s defensive systems. Further research is necessary to fully comprehend the mechanisms and limitations of the defense against parasitoids provided by this endosymbiont. Likewise, the defensive response of aphids infected with *Arsenophonus* can vary in quality based on the particular parasitoid species they encounter. These findings may only be applicable to the specific experimental conditions and insect species examined, and additional research may be necessary to validate these results.

## Figures and Tables

**Figure 1 insects-14-00763-f001:**
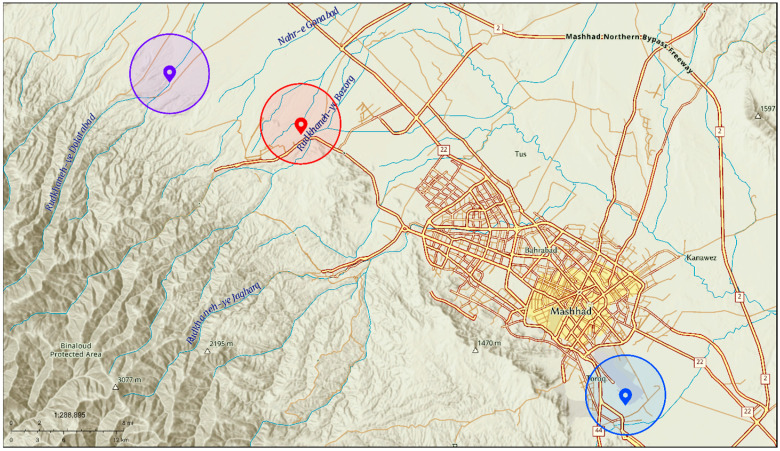
The three sampling sites of *Aphis craccivora* in the northeast region of Iran (Masshad, Razavi Khorasan Province). Blue = Toroq, 36°12′39″ N, 59°39′01″ E, 1007 m; Red = Hesar-e Sorkh, 36°24′52″ N, 59°20′38″ E, 1250 m; and Purple = Kahu, 36°27′08″ N, 59°13′30″ E, 1370 m.

**Figure 2 insects-14-00763-f002:**
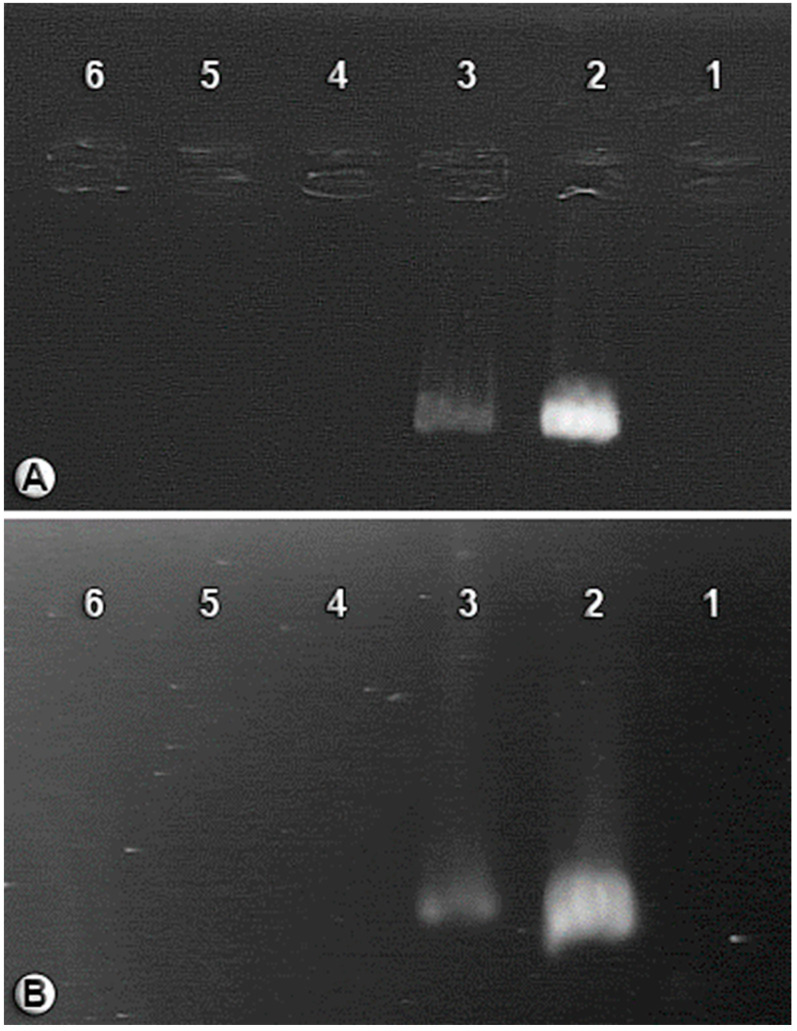
Detection of the presence of *Arsenophonus* sp. via PCR (**A**) fbaA gene and (**B**) ftsK gene. (1) Negative control; (2) Control Group (CG), *Aphis craccivora* infected with *Arsenophonus* (positive control); (3) *Arsenophonus*-reduced infection (AR) with the initial round of antibiotic; and (4–6) *Arsenophonus*-reduced infection (AR) with the second round of antibiotic.

**Figure 5 insects-14-00763-f005:**
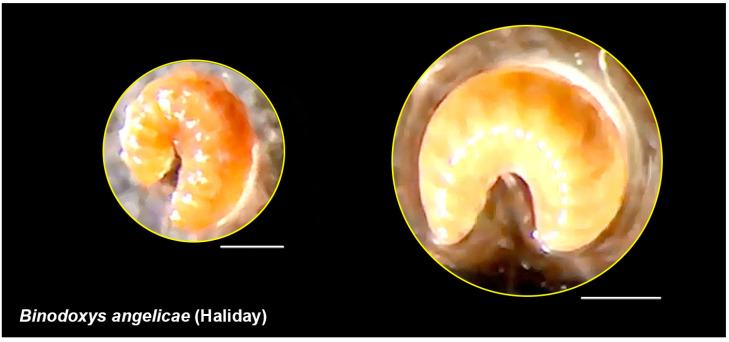
Effect of the endosymbiont bacterium *Arsenophonus* on the development of *Binodoxys angelicae* (Aphidiiniae: Braconidae). Left: a positive effect as the egg hatches and reaches the last larval stage. Right: a negative effect as the egg ceases developing at the larval stage (these observations were made during the experiment). Scale bar = 500 μm.

## Data Availability

The data presented in this study are available from the corresponding authors upon reasonable request.

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
