# Peer review of "Arsenophonus: A Double-Edged Sword of Aphid Defense against Parasitoids"

_insects, 2023, doi:10.3390/insects14090763_

Round 1
Reviewer 1 Report
The authors appear to be addressing a small gap in knowledge about this well-known insect host-microbe interaction. A similar study has been performed, but did not take the bioassay of the parasitoid out far enough to be able to make any observations about the wasp larvae development within the host.
The ms appears to be well written with proper experimental design, execution, and interpretation. I cannot speak to the molecular methods or interpretation.
In Fig 2 and Fig 3, perhaps include the stats in the caption or with asterisks in the graphs?
Why not cite J Asia-Pacific Entomol 2017 20:794-801 review?
Also, perhaps clarify that the Wulff et al PLoS One 2013 paper did perform bioassay with a Binodoxys species, but did not look at larval development?
One item in the ms stands out, which the authors point out in the last two sentences in the Conclusions. It would have made a much better paper had the authors included more work on this. Addressing this would have made this study much more impactful.
Author Response
Dear the reviewer,
Thank you very much for reviewing our MS. We did all the corrections according to your comments and suggestions.
Please see the attached file.
Best regards,
Buntika and the co authors
Reviewer 2 Report
The manuscript entitled “Arsenophonus: a double-edged sword of aphid defense against parasitoids” is a very interesting work in which the authors shown the evidence of a symbiotic relation between a bacteria and species of aphids against their parasitoids. The results obtained are a reinforce to discuss this important biological interaction.
The manuscript in my view point is adequately introduced in all sections. The first section is complete and with all references for the support of ideas. Materials and methods are in order and clear. Also the results and discussion covered all the objectives proposed in the introduction. Only in the Fig. 5, I suggest to include the scale in which the photos are presented.
I do not have more comments, only to give to the authors my congratulations for this excellent work.
I appreciate a lot, also for your considerations to review this manuscript.
I think is good.
Author Response
Dear the Reviewer,
Please see the attached file.
Best,
Buntika and the co authors

Reviewer 3 Report
This study investigated the effects of an endosymbiont Arsenophonus on the parasitism rate of the black cowpea aphid by two parasitoid wasp species, Binodoxys angelicae and Lysiphlebus fabarum. The authors eliminated Arsenophonus by mixed antibiotics. They found that Arsenophonus did not defense the aphids against parasitizing by the wasps, but delay the emergence of the wasps and reduced the number of emergences. The authors conclude the Arsenophonus has indirect effects of protecting the aphids from being parasitized rather than direct effects. The quality of the article is not too bad, at least it can be understood.
Maijor concerns:
1. How to ensure that mixed antibiotics do not cure the obligate endosymbiont Buchnera ?
2. It is difficult to judge the curative effects of antibiotics on the endosymbiont only using the band of PCR products on the gel. Why not use quantitative PCR? Or at least the authors should provide photos of the gel.
Minor concerns:
1. The English expression is not clear enough. Such as L24-26 It is widely accepted that endosymbiont interactions with their hosts result in important effects, including the fitness of many pests and beneficial species. Beneficial associations are one particular type of endosymbiosis. I suggest having native speaker check the language expression.
2. Line 243 The nucleotide sequences deposited in the GenBank database. Please provide the Reference Number of the sequences
3. Line 267 (t= -0.94, f= 2.52, df= 78, P= 0.34). What is the “f” value? There are many similar issues in the following text.
4. Figure 2, 3 what are the vertical bars? SE or SD?
Author Response
Dear the reviewer,
Thank you very much for reviewing our MS. We did the corrections according to your comments and suggestions. Please see the attached file.
Best regards,
Buntika and the co authors.

Round 2
Reviewer 3 Report
(1) The author claimed that the error bars in Fig 3 and Fig 4 refer to standard error, but why are the upper and lower bars not the same length?
(2)Fig2 I suggest give the size of each band of the marker
Author Response
Dear the reviewer,
Please see the attached file.
Best,
Buntika and the co authors
